# Historical and projected future range sizes of the world's mammals, birds, and amphibians

Robert M. Beyer ●[1✉] & Andrea Manica ●[1]

Species' vulnerability to extinction is strongly impacted by their geographical range size. Formulating effective conservation strategies therefore requires a better understanding of how the ranges of the world's species have changed in the past, and how they will change under alternative future scenarios. Here, we use reconstructions of global land use and biomes since 1700, and 16 possible climatic and socio-economic scenarios until the year 2100, to map the habitat ranges of 16,919 mammal, bird, and amphibian species through time. We estimate that species have lost an average of 18% of their natural habitat range sizes thus far, and may lose up to 23% by 2100. Our data reveal that range losses have been increasing disproportionately in relation to the area of destroyed habitat, driven by a long-term increase of land use in tropical biodiversity hotspots. The outcomes of different future climate and land use trajectories for global habitat ranges vary drastically, providing important quantitative evidence for conservation planners and policy makers of the costs and benefits of alternative pathways for the future of global biodiversity.

---

[1] Department of Zoology, University of Cambridge, Downing Street, Cambridge CB2 3EJ, UK. ✉email: rb792@cam.ac.uk

Habitat range size is a strong predictor of species' vulnerability to extinction[1,2]. As a result, two major drivers of the decline of geographic range sizes—the conversion of natural vegetation to agricultural and urban land, and the transformation of suitable habitat caused by climate change—are considered two of the most important threats to global terrestrial biodiversity[3]. Land-use change has caused staggering levels of habitat contractions for a range of mammal[4–6], bird[7], and amphibian species[8]. Simultaneously, anthropogenic climate change has been driving shifts in species' ranges[9–12], which, whilst resulting in larger range sizes for some species, has led to severe range retractions for others[11,13,14]. Declines in global range sizes due to land-use and climate change heavily contribute to the loss of local species richness[15–17] and abundance[17–19] in many parts of the world, thereby threatening essential ecosystem functions[17,20]. With global agricultural area potentially increasing drastically in the coming decades[21], and climate change continuing to drive ecosystem change at an accelerating pace[22], future projections suggest that past trends in range contractions may continue[23,24], and likely contribute to projected large-scale faunal extinctions[12,13,25,26].

Considering the crucial role that species' range sizes play for extinction risks, a better understanding of the long-term range dynamics of individual species, and projections of future changes under alternative scenarios, is crucial for conservation planning from the local to the global scale. Such estimates would allow quantification of historical pressures on species, and inform prioritisation of future efforts. Here, we estimate the habitat range sizes of 16,919 mammal, bird and amphibian species from the year 1700 until 2100 based on global land use and climatic conditions. We use empirical datasets of the global distribution of species, and combine these with species-specific biome preferences to estimate local habitat suitability under natural vegetation, cropland, pasture and urban land cover. By overlaying these data with reconstructions of global biomes corresponding to past climatic conditions, and agricultural and urban areas since 1700, we estimate the historical habitat ranges of each species ('Methods'). We then extend the analysis into the future based on 16 alternative land use and climate trajectories until the year 2100, representing four emission scenarios (representative concentration pathways (RCPs) 2.6, 4.5, 6.0, 8.5), and five socio-economic pathways (shared socio-economic pathways (SSPs) 1–5) ('Methods'). SSP1 and SSP3 represent futures where socio-economic challenges for adaptation and mitigation to climate change are both low and both high, respectively; SSP4 combines high challenges to adaptation with low challenges to mitigation, while SSP5 represents the opposite case; SSP2 is a middle-of-the-road scenario of intermediate challenges to adaptation and mitigation[27,28] RCPs 2.6–8.5 represent increasing levels of global warming by the end of the century[29]. Considering all possible SSPs for any given RCP is crucial, as using only one realisation per RCP can conflate effects and lead to contestable patterns (e.g., ref. [16]). By design of the method used here, modelled species' habitat ranges do not exceed the outermost geographic limits of species' observed and projected occurrences. Whilst this approach still allows for ample range shifts and expansions ('Methods'), climate change may push some species beyond these bounds, which our estimates would not account for. Furthermore, our method does not account for habitat range shifts from climatic changes that are too small to manifest as biome changes ('Methods'); thus, range shifts in highly climatically sensitive species may be underdetected. Our estimates of the distribution of species' habitat ranges based on land use and climate represent upper estimates for the actual distribution of populations. They neither incorporate other types of human influence, such as hunting[30], suppression by introduced species[31] and pathogens[32],

nor do they account for species' mobility[33], or the impacts of habitat fragmentation[34] and trophic cascade effects[35] on the viability of local populations. Our analysis reveals that species have lost an average of 18% of their natural range sizes thus far, a figure that may drop to 13% or increase to 23% by the end of the century, depending on future global climatic and socioeconomic developments.

## Results and discussion

**Historical changes in habitat range sizes**. With moderate impacts on the habitat ranges of most species' up until the industrial revolution, the expansion of agricultural production and settlements alongside the rise in population growth since the early 1800 s has drastically reduced range sizes of most mammals, birds, and amphibians (Fig. 1a). Using potential natural ranges in 1850 as a reference ('Methods'), we estimate that species had lost an average of 18% of their natural habitat area by 2016. For most species, alterations in the global distribution of biomes due to past climatic change have had a much smaller effect on range sizes compared to land use, causing average range changes of <1% in the past 300 years (Supplementary Fig. 1). There is substantial variability between species in terms of the experienced range changes. Critical levels of habitat range loss affect a rapidly rising number of species, with currently 16% have lost more than half of their natural range. Among these species, tropical species account for an increasingly larger proportion (Fig. 1b), whereas small-ranged and threatened species did not experience significantly higher ranger losses than other species (Supplementary Fig. 2). For an estimated 18% of species, ranges have expanded in consequence of anthropogenic climate change and the conversion of unsuitable natural vegetation to cropland and pastures (Fig. 1a).

The magnitude of habitat range contractions estimated since 1700 is not merely the result of the increasing area of converted land. Over recent centuries, range loss has increased disproportionately in relation to the total size of agricultural and urban areas (Fig. 2a). Whilst the first billion hectares converted since 1700 caused an average 3% loss of habitat size, the most recently converted half billion hectares are responsible for an average loss of 6% of natural range sizes. This acceleration of marginal range losses can be explained by a long-term trend in the location of land-use change towards tropical regions, where both local species richness is higher and average ranges sizes are smaller, and thus where the destruction of natural habitat leads to particularly high relative range losses[36] (Fig. 2b, c). Following a long period of much less land conversion than in other parts of the world, these areas have experienced a rapid expansion of agriculture since the end of the 19th century. Habitat conversion rates reached their highest levels to date in South America around the mid–late 20th century, and in the late 20th and early 21st century in South East Asia (Fig. 2b), a global hotspot of small-ranged species[36] (Fig. 2c).

**Projected future changes in habitat range sizes**. Whether these past trends in habitat range losses will reverse, continue or accelerate will depend on the global emission and socio-economic pathway chosen in the coming years and decades. By 2100, average range losses could reach up to 23% in the worst-case scenario (RCP 6.0, SSP 3), or drop to 13%—roughly equivalent to levels in 1955—in the best case (RCP 2.6, SSP 1) (Fig. 3a). The proportion of species suffering the loss of at least half of their natural range size could increase to 26% (RCP 6.5, SSP 3) or decrease to 14% (RCP 2.6, SSP 1) by 2100 (Fig. 3b). Isolating the impact of climate change shows that higher levels of global warming increase both the number of species experiencing substantial range contractions and range expansions[11,14]

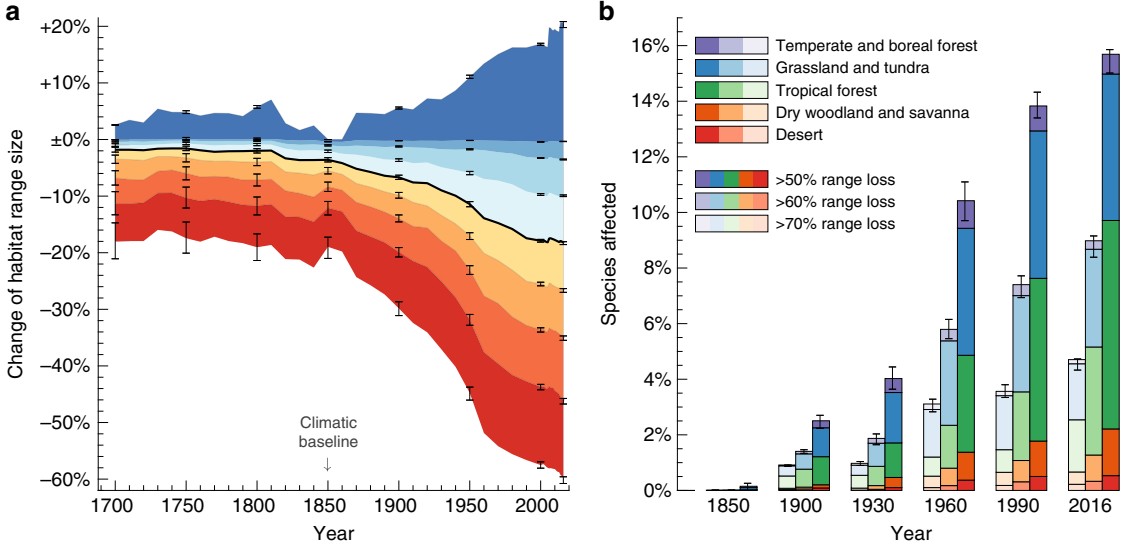

**Fig. 1 Past changes in the range sizes of mammals, birds and amphibians. a** Estimated changes in range sizes between 1700 and 2016, relative to potential natural ranges in 1850 (which are based only on the natural biome distribution and assume no land use; 'Methods'). Coloured areas represent 10%, 20%, ..., 90% percentiles (red, orange, ..., blue) of range changes across 16,919 species; the black line shows the across-species median. **b** Percentages of species affected by critical range contractions, broken down by species' primary mega-biome ('Methods'). Upper and lower uncertainty bars of the percentiles in (**a**) and the critical losses in (**b**) correspond, respectively, to the same analyses based on the available upper and lower uncertainty bounds of the land-use reconstruction ('Methods').

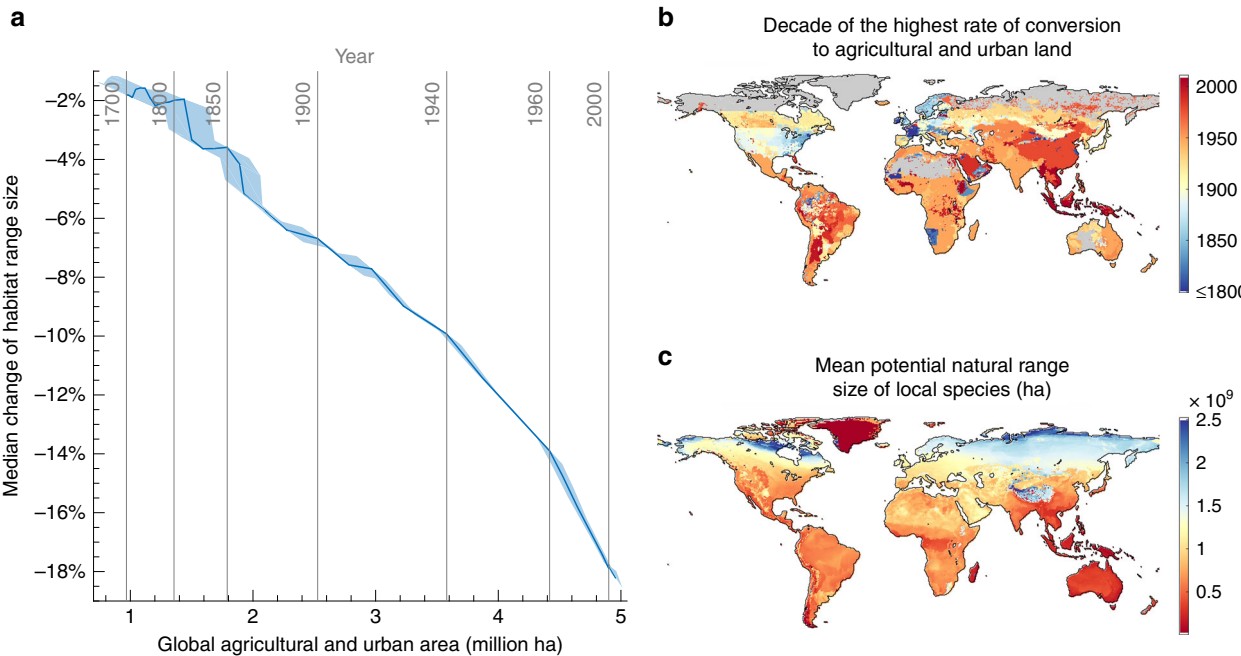

**Fig. 2 Acceleration of the marginal impact of land use on species' range sizes. a** Across-species median range loss against the cumulative global agricultural and urban area. Uncertainty bands are based on the uncertainty of the land-use reconstruction. **b** Local time of the highest rate of conversion to agricultural and urban land up until 2010. **c** Mean potential natural range size of species locally present for the natural biome distribution in 1850 and no anthropogenic land use.

(Supplementary Fig. 1). Across-species average range losses by 2100 increase consistently with higher emission levels for any given socio-economic pathway (Fig. 3a). At the same time, the differences between climate-change scenarios, in terms of average range change, are at times smaller than the differences between socio-economic scenarios. Across climate change scenarios, average range loss is consistently highest for SSP 3 (high challenges for both mitigation and adaptation to climate change), similar for SSP4 (adaptation challenges dominate), SSP5

(mitigation challenges dominate) and SSP2 (intermediate challenges), and lowest for SSP1 (low challenges for both mitigation and adaptation). Whilst SSP 1 would enable the re-expansion of ranges in many parts of the world as the result of the abandonment of agricultural areas, notably in Southeast Asia, SSP 3 represents a continuation of land-use change in the tropics, most strongly in the Congo basin (Supplementary Fig. 4).

Our estimates of the past and present states of species' habitat ranges, and how they will be impacted under alternative future

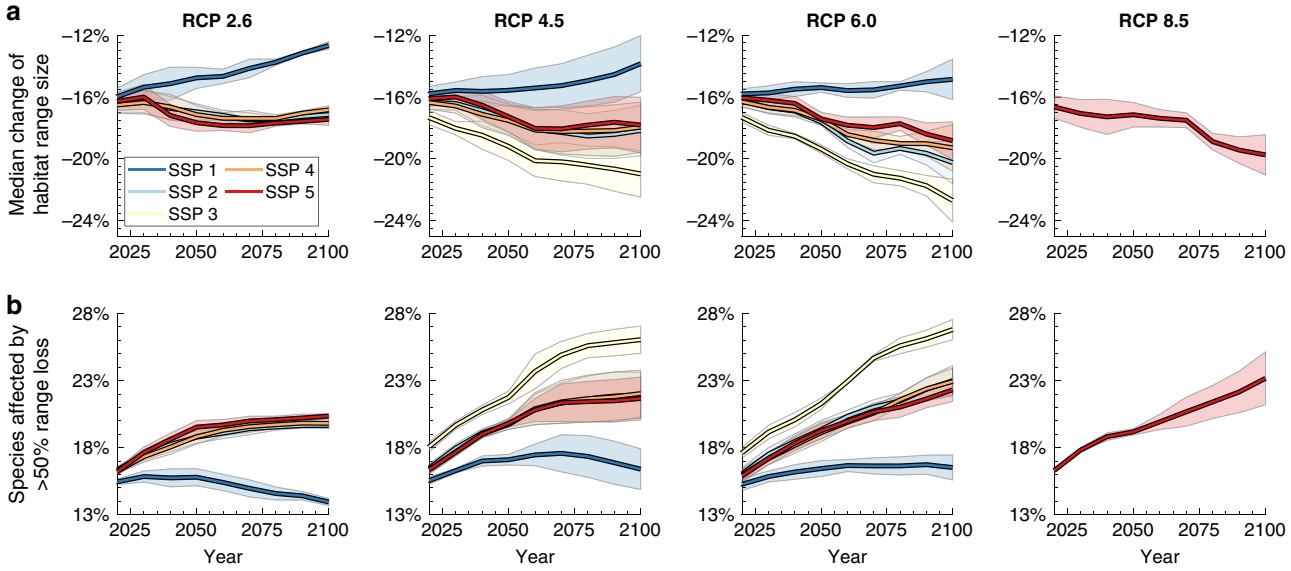

**Fig. 3 Projected future range changes of mammals, birds and amphibians for representative concentration pathways (RCPs) 2.6, 4.5, 6.0, 8.5, and shared socioeconomic pathways (SSPs) 1–5. a** Across-species median range size changes, relative to potential natural ranges in 1850 (analogous to the black line in Fig. 1a). **b** Percentage of species projected to experience a loss of more than half their natural range size. In (**a**), (**b**), lines represent the means of projections derived based on n = 3 different climate models; uncertainty bands represent standard deviations and indicate the uncertainty of the projections with respect to the climate data ('Methods'). Uncertainties of the AIM land-use projections of specific RCP/SSP scenarios are not available. RCP/SSP scenarios not shown are either incompatible or the relevant land-use data were not generated with the AIM model[51]. Complete fan charts showing 10%–90% percentiles of range changes across species (as in Fig. 1a), and bar charts showing critical range losses by primary mega-biome (as in Fig. 1b), are shown in Supplementary Fig. 3 for individual RCP/SSP combinations.

climatic and socio-economic scenarios, provide important evidence for conservation-oriented decision-making from the local to the global scale. Our results provide quantitative support for policy measures aiming at curtailing the global area of agricultural land[37,38] (by sustainably intensifying production[39–41], encouraging dietary shifts[42,43] and stabilising population growth[44]), especially in areas of small-range species[36], steering production to agro-ecologically optimal areas when the additional expansion is inevitable[39,45], targeting land abandonment and restoration in hotspot areas[46,47] and limiting climate change[48]. Whilst our data quantify the drastic consequences for species' ranges if global land use and climate change are left unchecked, they also demonstrate the tremendous potential of timely and concerted policy action for halting and indeed partially reversing previous trends in global range contractions.

## Methods

**Global land-use data**. For the historical time period 1700–2016, we used reconstructions of global cropland, pasture, and urban areas from the HYDE 3.2 dataset[49] (available from https://doi.org/10.17026/dans-25g-gez3). Whilst HYDE 3.2 provides land-use data as far back as 10,000 BCE, we began our analysis in the year 1700, prior to which global land-use data are subject to increased uncertainty[49,50]. A total of 47 maps, including lower and upper uncertainty bounds, are available at 10-year intervals between 1700 and 2000, and at 1-year intervals between 2000 and 2016. These data were upscaled from their original spatial resolution of 0.083° to a 0.5° grid by summing up the cropland, pasture, or urban areas of all 0.083° grid cells contained in a given 0.5° cell.

For the period 2020–2100, we used 0.5°-resolution 10-year time-step projections of global cropland, pasture, and urban areas from the AIM model[51] (available from https://doi.org/10.7910/DVN/4NVGWA), covering Representative Concentration Pathways (RCPs) 2.6, 4.5, 6.0 and 8.5, and Shared Socio-economic Pathways (SSPs) 1–5. The dataset contains all possible combinations of these emission and socio-economic trajectories with the exception of RCP 2.6/SSP 3, and RCP 8.5/SSPs 1–4. The data were harmonised with the HYDE 3.2 data by adding the differences between HYDE 3.2 and AIM cropland, pasture and urban area maps in the year 2010 to the AIM future land use projections. We refer to refs. [27–29,52] for details of the emission and socio-economic pathways, and to ref. [28] for a comparison between the AIM model and other integrated assessment models.

**Global biome data**. We used the BIOME4 vegetation model[53] (available from https://pmip2.lsce.ipsl.fr/synth/biome4.shtml) to simulate the distribution of global potential natural biomes between the years 1700 and 2000, and between 2020 and 2100 for each of the four climate-change scenarios considered here (RCPs 2.6, 4.5, 6.0, 8.5), at a spatial resolution of 0.5°. Inputs required by BIOME4 include global mean atmospheric $CO_2$ concentration, and gridded monthly means of temperature, precipitation, and percent sunshine. Past and RCP-specific future $CO_2$ levels were obtained from refs. [54] and [55], respectively. The climatic data were generated as follows. For the period 1700–1900, we used annual simulations from the HadCM3 climate model[56] (available from https://esgf-node.llnl.gov/search/cmip5/; Experiments 'past1000' and 'historical', Ensemble 'r1i1p1'). For the period 1901–2016, we used 0.5° resolution annual observational data[57] (available from https://doi.org/10.5285/10d3e3640f004c578403419aac167d82). For the period 2020–2100, and for each RCP (2.6, 4.5, 6.0, 8.5), we used annual simulations from the HadGEM2-ES climate model[58], the MIROC5 climate model[59] and the CSIRO-Mk3.6.0 climate model[60] (available from https://esgf-node.llnl.gov/search/cmip5/; for each climate model and each RCP, we used averages from Ensembles 'r1i1p1', 'r2i1p1', 'r3i1p1', 'r4i1p1'). We downscaled and bias-corrected both the pre-1901 HadCM3 simulations and the future HadGEM2-ES, MIROC5, and CSIRO-Mk3.6.0 simulations using the delta method[61]. This method is based on applying the difference between simulated and observed climate at times at which both are available (here we used the 1900–1930 period for the historical data, and the year 2006 for the future data) to the simulated climate at points in time at which only simulated data exist (i.e., pre-1901 and post-2016) in order to correct systematic biases in the climate model[61,62]. The delta method also serves to spatially downscale the simulated climate to the 0.5° resolution of the observational data.

For the computation of the global biome distribution at a point in time $t$, we used as inputs for BIOME4 the atmospheric $CO_2$ concentration and gridded monthly climate values averaged across the time interval $[t - 30$ years, $t]$. Biome simulations were performed at 10-year intervals for both the historical and the future period. The complete time series of global biome simulations are available as Supplementary Movies 1–13.

**Estimation of species' habitat ranges**. We estimated the geographic habitat ranges of an individual bird, mammal, and amphibian species through time following the general methodology in ref. [23]. Our approach combines the following data:

I. Spatial polygon data of species-specific extents of occurrence of all known birds[63] (available from http://datazone.birdlife.org/species/requestdis), mammals, and amphibians[64] (available from https://www.iucnredlist.org/).

II. Species-specific biome requirements[63,64] (data also available from the above websites).

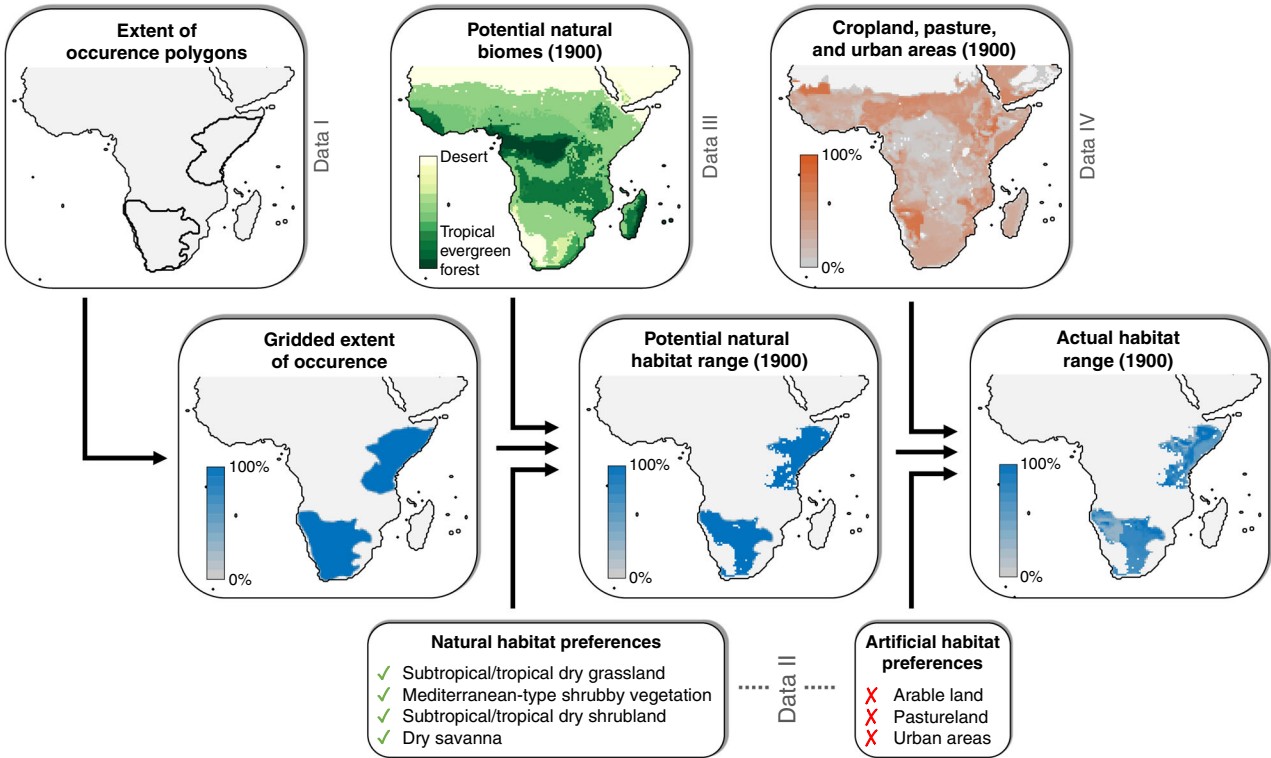

**Fig. 4 Method of estimating potential natural and actual range for the example of the bat-eared fox (*Otocyon megalotis*) in the year 1900.** Here, for visualisation purposes, cropland, pasture, and urban areas were aggregated into one map; in reality, our method checks each of them separately against species' artificial habitat preferences.

III. Maps of global potential natural biome distributions corresponding to the relevant climatic conditions through time (i.e., reconstructions for the past, and RCP-specific projections for the future).

IV. Maps of global cropland, pasture, and urban areas through time (i.e., reconstructions for the past, and RCP- and SSP-specific projections for the future).

The data I–IV were used to estimate the habitat range of individual species at a given point in time as illustrated in Fig. 4 and detailed in the following. In a first step, we used species-specific extents of occurrence (data I), which represent the outermost geographic limits of species' observed, inferred or projected occurrences[1]. These spatial envelopes do not account for the distribution of natural or artificial land cover within that area, and therefore generally extend substantially beyond a species' actual area of occupancy[65,66]. We first remapped extents of occurrence from their original spatial polygon format to a 0.083° resolution grid using the 'rasterise' function of the 'raster' package in R, which maps spatial polygons to those raster grid cells whose centres are contained within the polygons. For each species, we then determined the proportion of 0.083° cells contained in each 0.5° grid cell that represents the species' extent of occurrence. This provides an estimate of the proportion of each 0.5° grid cell that is contained in the species' extent of occurrence. Compared to the rasterising extent of occurrence directly to a 0.5° grid, this approach provides for more accurate estimates of species' ranges and reduces the number of species that are not included in our analysis because their extents of occurrence do not overlap with any grid cell centre.

In a second step, we refined the derived species-specific maps of the proportion of 0.5° grid cells contained in species' extents of occurrence by combining them with species-specific biome requirements and maps of global biome distributions. Species-specific biome requirements (data II) include one or more habitat categories (cf. Supplementary Table 1), in which each species is known to occur. A species was estimated as being present in a grid cell contained in its previously derived extent of occurrence under the potential natural biome at a given point in time if the species' list of habitat categories contained the local (i.e., grid cell-specific) potential natural biome at the relevant time (data III; see above). This required matching IUCN habitat categories (https://www.iucnredlist.org/resources/habitat-classification-scheme) with the biome categories of the Biome4 vegetation model, which was done as shown in Supplementary Table 1. In this way, we subset extents of occurrences by only retaining grid cells where the natural biome type is included in a species' list of suitable habitat categories. The result of this step represents a species' estimated potential natural habitat range (i.e., in the hypothetical absence of anthropogenic land use) at a given point in time.

In a third step, we estimated actual habitat ranges by including maps of global land use through time. Each species' actual habitat range at a given time was derived by removing any unsuitable anthropogenic land from the previously estimated potential natural range. Historical and projected future land use maps (data IV; see above) provide the fraction of each grid cell that is occupied by cropland, pasture or urban areas. These data were combined with information on which of these three artificial land cover types, if any, species can occur in, which is also included in the list of species' biome requirements (data II). This allowed us, for each grid cell contained in a species' potential natural range at a given time, to estimate the proportion of the grid cell that contained suitable habitat. A species' actual habitat range size was then obtained as the sum of the areas of the remaining suitable habitat from all relevant grid cells.

We applied the above method at each point in time for which global land use data is available (see above). In this way, we obtained potential natural ranges and actual ranges for 47 points in time between 1700 and 2016—using the baseline as well as lower and upper uncertainty bounds of the HYDE 3.2 land-use reconstructions—, and for nine points in time between 2020 and 2100—using the 16 combinations of future climatic and socio-economic pathways (see above), each of which, in turn, was considered based on climate data from three alternative models. Thus, we considered a total of 141 historical and 432 future scenarios.

Since the global distribution of natural biomes varies over time as the result of (naturally or anthropogenically) changing climatic conditions, the sizes of potential natural habitat ranges are time-dependent. This motivates to consider range changes in relation to the potential natural ranges estimated at a particular reference time, for which we chose the year $t_0 = 1850$, representing a modern pre-industrial baseline. Denoting the potential natural range and the actual range of a species $i$ at a time $t$ by $A_i^{\text{potential}}(t)$ and $A_i^{\text{actual}}(t)$, respectively, the range change associated with species $i$ at time $t$ as the result of the distribution of biomes and land use at that time was calculated at as

$$\Delta A_i(t) = 100\% \cdot \left( \frac{A_i^{\text{actual}}(t)}{A_i^{\text{potential}}(t_0)} - 1 \right). \tag{1}$$

Species whose potential natural habitat range size in the reference year $t_0 = 1850$ (i.e., the range size estimated in the absence of anthropogenic land use and based on the global distribution of biomes in 1850) is zero, $A_i^{\text{potential}}(t_0) = 0$, were not included in the analysis as, in this case, changes in range size are not defined. Based on the set $\{\Delta A_i(t)\}_{i=1,2,\dots}$ of the individual range changes of all species through time, we calculated range change percentiles at each point in time (Fig. 1a), and determined the proportion of species that have experienced the loss of a given

percentage of their baseline range (Fig. 1b). Similarly as in Eq. (1), we also computed the range change attributed only to climate-change-induced biome changes, $100\% \cdot \left( A_i^{\text{potential}}(t)/A_i^{\text{potential}}(t_0) - 1 \right)$ (Supplementary Fig. 1).

Analyses were conducted using Matlab R2019a[67] and R v3.6.3[68].

**Method discussion**. Whilst the available climate data for a given point in time only allows us to assign one primary natural biome type to each 0.5° grid cell, microclimates within cells may, in reality, result in the presence of different biomes in parts of a cell that are not represented in our data. By design of the approach used here, grid cells containing a non-primary biome that is suitable for a species, whilst the estimated primary biome is not, do not contribute to our estimation of the species' habitat range. Conversely, grid cells containing a non-primary biome that is not suitable for a species, whilst the primary biome is suitable, would be included in their entirety in the species' estimated range. This may lead us to underestimate the range sizes of species typically occurring in non-primary biomes in areas in which the estimated primary biomes are not suitable for the species, and to overestimate the range sizes of species typically occurring in the estimated primary biome in areas where other biomes also occur that are not suitable. Higher-resolution biome data could, in principle, reduce inaccuracies; however, generating such data in a reliable manner is not trivial. We are not aware of indications that this aspect of the approach would either systematically increase or decrease our overall estimates for range size changes across species in Fig. 1a.

Our estimation of species' habitat range sizes does not take into account habitat connectivity within or across grid cells. In principle, this can result in disconnected patches being included in a species' estimated range, despite in reality being too small to represent potentially suitable habitat. However, neither species-specific data on the minimum size that spatially connected areas must not fall below before becoming non-viable nor reliable very-high-resolution land use and biome data, both of which would be needed to fully accommodate this issue, are currently available.

Although species' extents of occurrence are based not only on known, but also inferred and projected occurrences, the data remain very likely biased as the result of range contractions that occurred before the beginning of the systematic collection and mapping of species' distributions, and that cannot be fully reconstructed. Whilst this may lead us to underestimate the absolute range sizes of species, it does not necessarily imply that we either systematically underestimate or overestimate the percentage change of species' ranges through time.

We chose the 0.5° resolution for our analysis as both the 1901–2016 observational climate data (and therefore also the pre-1901 and future climate data, which were downscaled using the observational data) and the projections of future land use are only available at this resolution. Attempts to further downscale these data would likely involve significant additional uncertainties. We are not aware of indications that an increase in the resolution of the analysis (if indeed the necessary datasets were available) would result in a systematic increase or decrease of either the absolute range sizes or the percentage change of range sizes relative to the baseline sizes, estimated here, at any point in time.

Species-specific extents of occurrence and habitat preferences have been argued to be subject to uncertainty[69]; however, uncertainty estimates (quantitative or otherwise) are not provided with the data. In our main analysis, we therefore used the available data at face value. However, to verify that our results are not overly impacted by specific species, we performed the following bootstrapping analysis. Based on the set of species-specific range changes of all 16,919 species, estimated for the year 2016, we randomly sampled 16,919 values from this set with replacement a total of $10^4$ times. For each of these $10^4$ sets of range change estimates, we calculated 10%–90% percentiles analogous to Fig. 1a. For each percentile, we then calculated the mean and standard deviation of the computed $10^4$ values. The result, shown in Supplementary Fig. 5, demonstrates that the uncertainties of our estimates with respect to specific species are very small, indicating that our results are robust with respect to potential uncertainties in the species data.

Estimates of temporal delays in biome shifts in response to climatic changes[70] are currently not available with the global coverage that would allow us to further refine our approach of assuming that biomes at a given point in time are determined by the climatic conditions in the preceding 30 years. This also applies to data on the dispersal speeds of plant functional types, and their effect on potential delays in colonisations of previously climatically unsuitable areas[33]; current studies on this topic are too spatially scarce to inform our approach. In our main analysis, we therefore followed the assumption commonly made in global vegetation models of no seed dispersal limitations[71]. However, to explore the impact of this assumption, we also repeated our analysis based on the extreme scenario of biomes not shifting at all between the present (year 2016) and 2100. The estimated range size changes (Supplementary Fig. 6) are quantitatively similar to the results of our main analysis (Fig. 3), consistent with our assessment of the overall stronger impact of land use compared to climate-driven biome changes. Qualitatively, i.e., in terms of how different RCP/SSP scenarios rank relative to each other, results are equivalent to those of our main analysis.

As noted in the Introduction, our estimates of future habitat ranges represent upper estimates of species' actual geographic distributions. In particular, our main analysis does not account for species' ability to migrate to areas that will become suitable habitat at a future point in time but are not at present. However, our framework allows us to examine the effect of excluding such areas from the estimated habitat range. We repeated our analysis of future changes in habitat range sizes, but considered a grid cell as part of a species' range only if the local biomes estimated for both the relevant point in the future and for the present (year 2016) were included in the species' list of biome requirements. In other words, grid cells outside of species' current potential natural habitat ranges were not counted towards their future range sizes, assuming that species are not able to migrate at all. This represents an extreme scenario that will underestimate most species' mobility (e.g., over half of the species considered here can fly) and their ability to track biome shifts. Since the habitat range derived for a species in this manner is a subset of the one estimated in our main analysis, projected range losses based on this approach are, by design, higher (Supplementary Fig. 7). Qualitatively, results are equivalent to those in Fig. 3 in terms of how different RCP/SSP scenarios rank relative to each other.

As the empirical data on species' habitat preferences only provide categorical biome requirements, not continuous climatic envelopes, the method used here does not account for range changes due to changes in climatic conditions that are too small to manifest as biome changes. However, estimating precise climatic envelopes of species can be subject to considerable uncertainty and be highly sensitive to the way in which they are estimated (see below). By construction of the method used here, species' ranges over time vary within the extents of occurrence provided with the empirical data, and do not exceed those. Justification for this assumption is provided by the fact that potential natural ranges (and, much more, actual ranges) are generally well-contained within extents of occurrence, with the former accounting for an average of 64% of the area of the latter in the reference year 1850, thus providing ample space for range shifts and expansions within the boundaries. Additional evidence that the restriction of habitat ranges to the extents of occurrence does not prevent significant range expansions can be seen in the sizeable number of species that have already experienced such range expansions (Fig. 1a and Supplementary Fig. 1) or are predicted to do so in future scenarios of strong global warming (Supplementary Fig. 1 and Supplementary Fig. 3a).

Climate niche models estimate statistical relationships between climatic conditions and species' spatial distributions, and apply these to climate projections[72] in order to estimate future distribution patterns. By design, they have great potential for mapping species' distributions under a high degree of complexity in terms of possible predictor variables and their interactions, which has made the approach very useful in scenarios where the number of species, the geographic region and/or the temporal scale considered is relatively small so that statistical challenges are well-manageable[73–75]. In an analysis involving a large number of species, points in time, and different climatic and land-use scenarios considered here, the challenges commonly faced by climate nice models, specifically in terms of ensuring the robustness of the underlying statistical model and the estimated parameters, and avoiding unwanted artefacts in the extrapolation behaviour[76–81], would be very difficult to manage. By operating directly and transparently on the empirical data of species' extents of occurrence and biome requirements, and not being reliant on any particular statistical model or parameterisation, the approach used here provides the robustness needed at this scale of data[23,82].

**Reporting summary**. Further information on research design is available in the Nature Research Reporting Summary linked to this article.

## Data availability
Data associated with this study are available on the Open Science Framework https://doi.org/10.17605/OSF.IO/TJ6C5. Bird distribution and habitat data are available from http://datazone.birdlife.org/species/requestdis; mammal and amphibian distribution and habitat data are available from https://www.iucnredlist.org/; HYDE 3.2 past land use reconstructions are available from https://doi.org/10.17026/dans-25g-gez3; AIM future land-use projections are available from https://doi.org/10.7910/DVN/4NVGWA; Historical and future climate simulations are available from https://esgf-node.llnl.gov/search/cmip5/; 1901–2016 observed climate data are available from https://doi.org/10.5285/10d3e3640f004c578403419aac167d82.

## Code availability
Code associated with this study is available on the Open Science Framework https://doi.org/10.17605/OSF.IO/TJ6C5.

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

## Acknowledgements

The authors are grateful to BirdLife International for sharing and advising on the bird species distribution data, and to Mario Krapp for advising on the climate data. R.M.B. and A.M. were supported by ERC Consolidator Grant 647797 'LocalAdaptation'.

## Author contributions

R.M.B. conducted the analysis and wrote the paper. R.M.B. and A.M. designed the project, interpreted the results and revised the paper.

## Competing interests

The authors declare no competing interests.
