## [Peer Review File · Nature Communications]

This manuscript has been previously reviewed at another journal that is not operating a transparent peer review scheme. This document only contains reviewer comments and rebuttal letters for versions considered at *Nature Communications*.

Reviewers' Comments:

Reviewer #1:

Remarks to the Author:

The authors have made substantial improvements to the paper for which I am grateful and thank them for their efforts. However, there remains one major problem with the methodology. In projecting future climates the authors use only a single global climate model. There is substantial difference in climatic projections between different global climate models. The state-of-the-art in climate change impact modelling is to consider multiple global climate model projections as an ensemble. I would still not be prepared to consider the paper for publication in a journal such as *Nature Communications* based on the use of a single global climate model due to the sensitivity of the results and the implications to model specific climatic projections. The authors need to consider multiple global climate model projections in their analyses and present the uncertainty derived from these projections. I would be prepared to review the paper again once this methodological flaw has been addressed.

Reviewer #2:

Remarks to the Author:

This manuscript presents an analysis of the changes in the geographic ranges of a range of animal species from the past to the future. The authors conclude that while future changes in animal species ranges could be larger than anything that happened since the mid-Holocene, different future trajectories of climate and land use change could lead to very different outcomes, thus providing further impetus for international measures to mitigate climate and land use change in the future. In general this study is interesting, although given the many uncertainties, it strikes a rather overconfident message about species range change, particularly with respect to temporal lags in biome change, the potential for some species to adapt, and the existence persistence of habitats at spatial scales finer than the rather coarse 0.5° resolution at which the study was performed. Furthermore, the paleo-analysis seems largely superfluous and the source of much uncertainty. For these reasons, I am not particularly comfortable about recommending this manuscript for publication in *Nature Communications*, but after substantial revision, it might be acceptable.

This study stands on its own without a consideration of the paleo-perspective. The authors do not make a convincing argument as to why the Holocene analysis is valuable, and given the large degree of uncertainty surrounding past climate and land use, this element of the study distracts from the main message about the influence of future climate and land use trajectories on species ranges and potential for extinctions. No climate model currently reproduces mid-Holocene climate in a way that is consistent with independent paleoclimate reconstructions (e.g., Ackerley et al., 2017; Braconnot et al., 2007; Chevalier et al., 2017; Lin et al., 2019; Mauri et al., 2014). The authors' assurances that the HYDE scenario is widely used, that does not justify its suitability or accuracy as a reconstruction of past land use. The authors explicitly ignore historical anthropogenic impacts on species ranges that are not caused by conversion to agricultural land or pasture or urban areas. Non-agricultural land use, including hunting, fire, introduction of exotic invasives have been demonstrated to have had major impacts on species ranges (e.g., Boivin et al., 2016). As the climate and agricultural land use fluctuations are relatively small between 7ka and 1850, these other drivers could be very important. In conclusion, this study would be simpler, easier to justify and rationalize given uncertainties farther back in the past, and its message would have more impact if the authors simply considered the period from 1850 to 2100 CE. I encourage the authors to present the paleo-aspect of their work in a separate publication, where they could further explore uncertainties in paleoclimate and past land use, and alternative drivers of species range changes besides land conversion.

Specific comments:

Figure 1A

The 1850 baseline should be indicated on the plot, and it would be helpful to explain why the values are not zero at this time (because of land use changes that already occurred relative to potential natural by 1850 CE).

Line 124-125

It appears that the caption and the figures B and C are swapped. There seem to be some strange artifacts in Fig. 2B, e.g., outline of Namibia, Saudi Arabia. How is it possible that India and Mexico both have the decade of highest land conversion rate sometime in the 20th century when these countries have had millennia of land use? This figure would not be necessary or simpler to justify if the authors limited the study to the post-1850 period.

Line 191

HYDE is an acronym and should be capitalized throughout the manuscript

Lines 198-200

Just because HYDE is widely used doesn't mean it is correct. Limiting the study to the Industrial Era and future would avoid this problem. Otherwise, the authors need to provide further evidence supporting the appropriateness of the HYDE scenarios, particularly for influencing past species ranges, and why other impacts on species ranges besides land conversion could be ignored.

Line 205

The appropriateness of using 0.5° spatial resolution to estimate species ranges should be discussed/justified. Wouldn't this depend on the type of animals? How is connectivity/continuity of habitats represented at this scale?

Line 214

BIOME4 is a model name and should be capitalized throughout the manuscript

Line 245

The color scheme for biomes in the animation is confusing and it is not appropriate to use a continuous color scale to represent categorical data. A standard color scheme for BIOME4 biomes is available and should be used <https://pmip2.lsce.ipsl.fr/synth/biome4.shtml>.

Line 302-305

Following up on my comment above, spatial resolution and microhabitat is mentioned here, but the effects that these might have on the results are not discussed. Some further discussion and quantification of the uncertainties would be helpful.

References

- Ackerley, D., Reeves, J., Barr, C., Bostock, H., Fitzsimmons, K., Fletcher, M.-S., Gouramanis, C., McGregor, H., Mooney, S., Phipps, S. J., Tibby, J., & Tyler, J. (2017). Evaluation of PMIP2 and PMIP3 simulations of mid-Holocene climate in the Indo-Pacific, Australasian and Southern Ocean regions. *Clim Past*, 13(11), 1661-1684. doi:10.5194/cp-13-1661-2017
- Boivin, N. L., Zeder, M. A., Fuller, D. Q., Crowther, A., Larson, G., Erlandson, J. M., Denham, T., & Petraglia, M. D. (2016). Ecological consequences of human niche construction: Examining long-term anthropogenic shaping of global species distributions. *Proc Natl Acad Sci U S A*, 113(23), 6388-6396. doi:10.1073/pnas.1525200113
- Braconnot, P., Otto-Bliesner, B., Harrison, S., Joussaume, S., Peterchmitt, J. Y., Abe-Ouchi, A., Crucifix, M., Driesschaert, E., Fichet, T., Hewitt, C. D., Kageyama, M., Kitoh, A., Loutre, M. F., Marti, O., Merkel, U., Ramstein, G., Valdes, P., Weber, L., Yu, Y., & Zhao, Y. (2007). Results of PMIP2 coupled simulations of the Mid-Holocene and Last Glacial Maximum – Part 2: feedbacks with emphasis on the location of the ITCZ and mid- and high latitudes heat budget. *Clim Past*, 3(2), 279-296. doi:10.5194/cp-3-279-2007

Chevalier, M., Brewer, S., & Chase, B. M. (2017). Qualitative assessment of PMIP3 rainfall simulations across the eastern African monsoon domains during the mid-Holocene and the Last Glacial Maximum. *Quaternary Sci Rev*, 156, 107-120. doi:10.1016/j.quascirev.2016.11.028

Lin, Y., Ramstein, G., Wu, H., Rani, R., Braconnot, P., Kageyama, M., Li, Q., Luo, Y., Zhang, R., & Guo, Z. (2019). Mid-Holocene climate change over China: model-data discrepancy. *Clim Past*, 15(4), 1223-1249. doi:10.5194/cp-15-1223-2019

Mauri, A., Davis, B. A. S., Collins, P. M., & Kaplan, J. O. (2014). The influence of atmospheric circulation on the mid-Holocene climate of Europe: a data-model comparison. *Clim Past*, 10(5), 1925-1938. doi:10.5194/cp-10-1925-2014

Reviewer #3:

Remarks to the Author:

I previously reviewed this manuscript for Nature. The current version is much improved, with a much clearer description of the methods. Including shifts in biome distributions in the future projections makes the results more consistent. This study will make a useful and complementary addition to existing model projections of land-use impacts on biodiversity. I really like the demonstration of disproportionate impacts on narrow-ranged species, and the somewhat related disproportionate declines in the tropics. I still have some suggestions for improvements, mostly clarification of methods and assumptions, but perhaps also some additional consideration of uncertainty in the methods.

My main concern is that, if I have understood the methods correctly, species are not able to extend beyond their current estimated extent of occurrence, even in the future projections. This is a major assumption. The only justification given is that species currently occupy only part of their potential extent of occurrence, and so there is plenty of scope for range expansion. However, with rapid climate change in future, suitable areas for many species will move outside areas within current extent-of-occurrence maps. Indeed, substantial range shifts of species have already been observed. At the very least, this assumption needs to be made very clear, in the main text and not just in the methods section.

Another caveat of the study that needs to be made clear much earlier on is the omission of direct impacts of climate change on species distributions. The decision to consider climate effects mediated only via biome shifts is fine, and indeed may be the main mechanism for climate impacts on some species. However, many species are impacted directly by climate, and the decision not to try to model these impacts needs to be made clear very early in the paper. The introduction discusses climate impacts on species, but it is only in the Methods section that the limitation of considering only biome shifts is stated.

Uncertainty is considered in the historical reconstructions by calculating range change using the upper and lower bounds of estimated historical land use. No uncertainty is presented for the future projections, because the land-use projections do not have uncertainty bounds. No consideration is given to uncertainty in species' habitat preferences. I think this is probably OK, but certainly warrants some discussion in the paper. I do wonder though whether it would be possible to get at some estimate of the uncertainty associated with species habitat preferences (and there definitely is uncertainty in these!) using some sort of bootstrapping approach.

Specific comments:

Line 51: I think the claim that the methods can account "for the importance of interaction effects between" land use and climate change is a stretch. There are lots of ways that climate and land use interact (reviewed in Oliver & Morecroft, 2014, DOI: 10.1002/wcc.271), most of which cannot be captured by the methods used in this study.

Line 57-59: It would be helpful to explain here why all 20 combinations of RCP and SSP scenario were not considered.

Line 60: Should it be "adaptation" rather than "adaption"?

Lines 123-125: The maps in panels B and C seem to be described in the wrong order in the figure legend.

Line 132: So the proportion of species losing at least half their range increases from 2016 levels

even under the best-case scenario (RCP2.6, SSP1)? If so, it would be good to highlight this fact and explain why it happens despite the increase in average range area under the best-case scenario.

Lines 133-134: "increasing levels of global warming result in more species experiencing critical range losses". More compared with what? More as a function of increasing global warming? This is not very surprising. It would be nice to make some comparison between impacts of land conversion versus impacts of climate-driven biome shifts.

Lines 136-139: It is interesting that there was often more variability among SSP scenarios than among RCP scenarios. However, I can't help wondering whether this would still be the case if direct effects of climate change on species distributions had been captured by the models.

Line 140: It would be helpful to remind readers here what the different SSP scenario trajectories are.

Lines 146-149: It would be helpful in the figure legend to remind readers why some SSP-RCP combinations were not considered. Also, please explain in the legend why uncertainty was not considered in the future projections (assuming there is no way to do this).

Lines 155-156: Closing yield gaps may cause an increase in biodiversity. However, the land-use intensification that comes with closing yield gaps is typically associated with a reduction of biodiversity. Intensification effects on species were not included in the models here, so it is difficult to say with any confidence whether such a strategy would indeed be beneficial.

Lines 203-204: Give some more detail about the spatial upscaling methods used. E.g., what function was used to average values when upscaling?

Lines 229-230: More detail is needed describing the bias-correction method for the climate projections.

Lines 255-256: Give more detail about the species-specific biome types. Which data from Birdlife and IUCN were these exactly.

Lines 264-265: Give some more detail about the procedures used to convert spatial polygons to grid format.

Lines 279-280: The treatment of species land-cover preferences could be clearer here.

Lines 292-298: I think the treatment of baselines could be explained better. I think the gist of it is that the baseline 'potential' range is in the absence of human land use and for the distribution of biomes as in 1850. Is that correct?

Lines 362-363: The date of classification of threat status used in this figure needs to be given, as threat status changes over time.

Figure S3: It is really difficult to see the changes shown in this figure. I think it would be better to split this figure over multiple pages, so that the individual panels can be larger.

Lines 383-386: In the figure legend, explain why the estimates of total agricultural land for SSP5 for the different RCPs start at different levels in 2016.

Reviewer #4:

Remarks to the Author:

This is a fully-revised version of the paper I reviewed for Nature. I applaud the authors for the quality of the revision. I believe it was worth to go for Nat Comm that allows for longer and more detailed papers.

I only have few more comments that should not be too difficult to address:

- I think the authors must discuss that the habitat preferences they used are based on known current requirement/use, that already incorporate human influence on species ranges. Let's take an example with bears or wolves in Europe. If I was an expert, I would believe those species only prefer mountain habitats, sparsely-vegetated habitats etc... this is obviously not true. There are where the humans let them survive. This is likely to be true for hips of species, and I guess this is also very context dependent.

- While I appreciate the efforts of the authors to defend their modeling strategy, it has to remain fair and not misleading. The whole paragraph around climate niche model (L331-L345) it at best misleading. What is the point of using those papers very specific on few aspects and turn them as strong critics towards species distribution models? There are far more numerous papers demonstrating the usefulness of these models, comparison to independent data, simulated data etc... that show they do work remarkably well... It has been shown that ensemble modeling could address most of the pitfalls highlighted here, and last but not least, most of the caveats listed in

this section also applied to the approach used by the authors (extrapolation (see my previous comment), adequate data for calibration (this applies too here!)).
Wilfried Thuiller

We would like to thank the Reviewers again for their very helpful and constructive feedback, which has greatly helped us to improve our manuscript. We have incorporated all comments as detailed in the following point-by-point response.

Reviewer 1

The authors have made substantial improvements to the paper for which I am grateful and thank them for their efforts. However, there remains one major problem with the methodology. In projecting future climates the authors use only a single global climate model. There is substantial difference in climatic projections between different global climate models. The state-of-the-art in climate change impact modelling is to consider multiple global climate model projections as an ensemble. I would still not be prepared to consider the paper for publication in a journal such as Nature Communications based on the use of a single global climate model due to the sensitivity of the results and the implications to model specific climatic projections. The authors need to consider multiple global climate model projections in their analyses and present the uncertainty derived from these projections. I would be prepared to review the paper again once this methodological flaw has been addressed.

Following the Reviewer's suggestion, we repeated our analysis of the impacts of future climate change on species' ranges using ensemble-means from the CSIRO-Mk3.6.0 model and the MIROC5 model (in addition to ensemble-means from HadGEM2-ES that we had already used in our previous version). This has allowed us to quantify uncertainties with respect to the climate data to our projections in Fig. 3 and Fig. S1. Our main results have not changed as a result of including the additional climate data.

Reviewer 2

This manuscript presents an analysis of the changes in the geographic ranges of a range of animal species from the past to the future. The authors conclude that while future changes in animal species ranges could be larger than anything that happened since the mid-Holocene, different future trajectories of climate and land use change could lead to very different outcomes, thus providing further impetus for international measures to mitigate climate and land use change in the future. In general this study is interesting, although given the many uncertainties, it strikes a rather overconfident message about species range change, particularly with respect to temporal lags in biome change, the potential for some species to adapt, and the existence persistence of habitats at spatial scales finer than the rather coarse 0.5° resolution at which the study was performed. Furthermore, the paleo-analysis seems largely superfluous and the source of much uncertainty. For these reasons, I am not particularly comfortable about recommending this manuscript for publication in Nature Communications, but after substantial revision, it might be acceptable.

This study stands on its own without a consideration of the paleo-perspective. The authors do not make a convincing argument as to why the Holocene analysis is valuable, and given the large degree of uncertainty surrounding past climate and land use, this element of the study distracts from the main message about the influence of future climate and land use trajectories on species ranges and potential for extinctions. No climate model currently reproduces mid-Holocene climate in a way that is consistent with independent paleoclimate reconstructions (e.g., Ackerley et al., 2017; Braconnot et al., 2007; Chevalier et al., 2017; Lin et al., 2019; Mauri et al., 2014). The authors' assurances that the HYDE scenario is widely used, that does not justify its suitability or accuracy as a reconstruction of past land use. The authors explicitly ignore historical anthropogenic impacts on species ranges that are not caused by conversion to agricultural land or pasture or urban areas.

Non-agricultural land use, including hunting, fire, introduction of exotic invasives have been demonstrated to have had major impacts on species ranges (e.g., Boivin et al., 2016). As the climate

and agricultural land use fluctuations are relatively small between 7ka and 1850, these other drivers could be very important. In conclusion, this study would be simpler, easier to justify and rationalize given uncertainties farther back in the past, and its message would have more impact if the authors simply considered the period from 1850 to 2100 CE. I encourage the authors to present the paleo-aspect of their work in a separate publication, where they could further explore uncertainties in paleoclimate and past land use, and alternative drivers of species range changes besides land conversion.

Following the Reviewer's suggestion, we have removed the palaeo-perspective from our study. We begin our analysis slightly earlier than suggested by the Reviewer, at 1700 instead of 1850. We decided on this time frame because global population and land use data are subject to much less uncertainty from 1700 onward than before this point (Klein Goldewijk et al. 2017), and because we feel that Fig. 1 and 2 would lose a substantial part of their content if they started 150 years later.

In addition, for the period 1700-1900 climate (for which global observational data is not available), we now use downscaled and bias-corrected HadCM3 climate simulations available at annual time steps from the year 850 (from the CMIP5 database), instead of the previously used HadAM3H climate simulations which were only available at 1000-year time steps and needed to be temporally interpolated. Using the improved climate data has not changed our main results, which was to be expected given the relatively small climatic changes between 1700 and 1900.

Specific comments:

Figure 1A

The 1850 baseline should be indicated on the plot, and it would be helpful to explain why the values are not zero at this time (because of land use changes that already occurred relative to potential natural by 1850 CE).

We have indicated the 1850 baseline in the plot. We have also added the following clause to the caption:

... relative to potential natural ranges in 1850 (which are based only on the natural biome distribution and assume no land use; Methods)

Line 124-125

It appears that the caption and the figures B and C are swapped. There seem to be some strange artifacts in Fig. 2B, e.g., outline of Namibia, Saudi Arabia. How is it possible that India and Mexico both have the decade of highest land conversion rate sometime in the 20th century when these countries have had millennia of land use? This figure would not be necessary or simpler to justify if the authors limited the study to the post-1850 period.

The referee is correct about the order of subpanels in Fig. 2, and we apologise for the mistake. We have corrected the order in the caption.

We have changed "rate of land conversion" to "rate of conversion to agricultural and urban land" to clarify the type of land use. We have verified that the map has been generated correctly from the HYDE 3.2 dataset. The patterns shown appear plausible given the rapid increase in global agricultural and urban areas in the 19th and 20th century, compared to earlier points in time.

Line 191

HYDE is an acronym and should be capitalized throughout the manuscript

We have capitalised HYDE throughout the text.

Lines 198-200

Just because HYDE is widely used doesn't mean it is correct. Limiting the study to the Industrial Era and future would avoid this problem. Otherwise, the authors need to provide further evidence supporting the appropriateness of the HYDE scenarios, particularly for influencing past species ranges, and why other impacts on species ranges besides land conversion could be ignored.

We now begin the historical part of our analysis at 1700 as stated above. We have removed the statement referred to by the Reviewer, and have added a statement on the uncertainties in global land use data in the deeper past, as pointed out by the Reviewer.

Line 205

The appropriateness of using 0.5° spatial resolution to estimate species ranges should be discussed/justified. Wouldn't this depend on the type of animals? How is connectivity/continuity of habitats represented at this scale?

We have added the following paragraph to further explain the choice of the resolution of our analysis:

We chose the 0.5° resolution for our analysis, given that both the 1901–2016 observational climate data (which is also used to downscale and bias-correct both pre-1901 and future climate simulations) and the projections of future land use are only available at this resolution. Any attempt to further downscale these data would involve considerable additional uncertainties. We also note that we are not aware of any indication that an increase in the resolution of the analysis (if indeed the necessary datasets were available) would result either in a systematic increase or decrease, at any point in time, of both the absolute range sizes and the percentage change of range sizes relative to the baseline sizes, estimated here.

As stated in the main text, our approach does not account for habitat connectivity when estimating range sizes, i.e. each grid cell is treated independently. We have reiterated this in more detail in the Methods as follows:

Our estimation of species' range sizes does not take into account habitat connectivity within or across grid cells. In principle, this can result in disconnected patches being included in a species' estimated range despite in reality being too small to represent potentially suitable habitat. However, neither species-specific data on the minimum size that spatially connected areas must not fall below before becoming nonviable, nor reliable very-high-resolution land use and biome data, both of which would be needed to fully accommodate this issue, are currently available.

Furthermore, in our revised version, we have implemented a slightly improved method allowing us to quantify in a more precise manner the range size and percentage changes of species whose extent of occurrence (EEO) polygons overlap only with part of a 0.5° grid cell. In our previous version, such a cell would either entirely, or not at all, be counted towards a species' range – depending on whether (in addition to land cover-related criteria) the EEO polygons overlapped the centre of the 0.5° grid cell. In our revised version, we rasterised EEO polygons at a higher resolution of 0.083°, and then determined the proportion of 0.083° cells (representing EEO polygons) within each 0.5° grid cell. This has enabled us to move from a binary presence-absence measure to a [0,1]-interval continuous presence/absence measure on the 0.5° grid of our analysis. By design, the approach still allows us to conduct our main analysis on a 0.5° grid – which is sensible given the available climate and land use data, as

argued above –, while at the same time allowing us to estimate range sizes of species, whose EEO polygons cover certain 0.5° grid cells only partially, with a higher accuracy. We have changed the description of the rasterisation approach in the Methods accordingly.

Line 214

BIOME4 is a model name and should be capitalized throughout the manuscript

We have capitalised BIOME4 throughout the text.

Line 245

The color scheme for biomes in the animation is confusing and it is not appropriate to use a continuous color scale to represent categorical data. A standard color scheme for BIOME4 biomes is available and should be used <https://pmip2.lsce.ipsl.fr/synth/biome4.shtml> .

We now use a discrete palette based on maximally perceptually-distinct colours.

Line 302-305

Following up on my comment above, spatial resolution and microhabitat is mentioned here, but the effects that these might have on the results are not discussed. Some further discussion and quantification of the uncertainties would be helpful.

In addition to the changes made in response to the related query of the Reviewer further above, we have included the following statement in the Methods:

Whilst the available climate data for a certain point in time only allows us to assign one primary natural biome type to each 0.5° grid cell, microclimates within cells may in reality result in the presence of different biomes in parts of a cell that are not captured in our data. By design of our approach, grid cells containing a non-primary biome that is suitable for a species, whilst the estimated primary biome is not, do not contribute to our estimate of the species' range. Conversely, grid cells containing a non-primary biome that is not suitable for a species, whilst the primary biome is suitable, would be included in their entirety in the species' estimated range. This may lead us to underestimate the range sizes of species typically occurring in non-primary biomes in areas in which the estimated primary biomes are not suitable for the species, and to overestimate the range sizes of species typically occurring in the estimated primary biome in areas where other biomes also occur and are not suitable. Higher-resolution biome data could, in principle, reduce inaccuracies, however, generating such data in a reliable manner is not trivial. We also note that we are not aware of indications that this aspect of our approach would either systematically increase or decrease our overall estimates for range size changes across species in Fig. 1A.

Reviewer 3

I previously reviewed this manuscript for Nature. The current version is much improved, with a much clearer description of the methods. Including shifts in biome distributions in the future projections makes the results more consistent. This study will make a useful and complementary addition to existing model projections of land-use impacts on biodiversity. I really like the demonstration of disproportionate impacts on narrow-ranged species, and the somewhat related disproportionate declines in the tropics. I still have some suggestions for improvements, mostly clarification of methods and assumptions, but perhaps also some additional consideration of uncertainty in the methods.

My main concern is that, if I have understood the methods correctly, species are not able to extend beyond their current estimated extent of occurrence, even in the future projections. This is a major assumption. The only justification given is that species currently occupy only part of their potential extent of occurrence, and so there is plenty of scope for range expansion. However, with rapid climate change in future, suitable areas for many species will move outside areas within current extent-of-occurrence maps. Indeed, substantial range shifts of species have already been observed. At the very least, this assumption needs to be made very clear, in the main text and not just in the methods section.

Following the Reviewer's suggestion, we have added the following paragraph to the main text to summarise the information provided in more detail in the Methods:

By design of the method used here, modelled species' ranges do not exceed the outermost geographic limits of species' observed and projected occurrences. Whilst this approach still allows for ample range expansions and shifts (Methods), climate change may push some species beyond these bounds, which our estimates would not account for.

Another caveat of the study that needs to be made clear much earlier on is the omission of direct impacts of climate change on species distributions. The decision to consider climate effects mediated only via biome shifts is fine, and indeed may be the main mechanism for climate impacts on some species. However, many species are impacted directly by climate, and the decision not to try to model these impacts needs to be made clear very early in the paper. The introduction discusses climate impacts on species, but it is only in the Methods section that the limitation of considering only biome shifts is stated.

We have added the following summary of the relevant Methods section to the main text:

Our method does not account for range shifts from climatic changes that are too small to manifest as biome changes (Methods); thus, range shifts in highly climatically sensitive species may be underdetected.

Uncertainty is considered in the historical reconstructions by calculating range change using the upper and lower bounds of estimated historical land use. No uncertainty is presented for the future projections, because the land-use projections do not have uncertainty bounds. No consideration is given to uncertainty in species' habitat preferences. I think this is probably OK, but certainly warrants some discussion in the paper. I do wonder though whether it would be possible to get at some estimate of the uncertainty associated with species habitat preferences (and there definitely is uncertainty in these!) using some sort of bootstrapping approach.

We have conducted the bootstrapping approach suggested by the Reviewer, and have added the following statement to the paper:

Species-specific extents of occurrence and habitat preferences have been argued to be subject to uncertainty (Akçakaya et al. 2000); however, uncertainty estimates (quantitative or otherwise) are not provided with the data. We therefore used the available data at face value in our main analysis. However, to verify that our results are not overly impacted by specific species, we used the following bootstrapping approach. Based on the set of species-specific range changes of all 16,919 species, estimated for the year 2016, we randomly sampled 16,919 values from this set (with replacement) 10^5 times. For each of these 10^5 sets of range change estimates, we calculated 10-90% percentiles analogous to Fig. 1. For each percentile, we calculated

the mean and standard deviation of the available 10^5 values. The result, shown in Fig. S5, demonstrates that the uncertainties of our estimates with respect to specific species are indeed very small.

In addition, we have repeated our analysis of the impacts of future climate change on species ranges using ensemble-means from the IPSL-CM5A-LR and the MIROC-ESM-CHEM model (in addition to ensemble-means from HadGEM2-ES already used in our previous version). This has allowed us to quantify uncertainties of future estimates with respect to the climate data.

Specific comments:

Line 51: I think the claim that the methods can account “for the importance of interaction effects between” land use and climate change is a stretch. There are lots of ways that climate and land use interact (reviewed in Oliver & Morecroft, 2014, DOI: 10.1002/wcc.271), most of which cannot be captured by the methods used in this study.

We have removed the sentence pointed out by the Reviewer.

Line 57-59: It would be helpful to explain here why all 20 combinations of RCP and SSP scenario were not considered.

Fujimori et al. (2018) state in their paper describing their land use dataset that other scenarios “are either incompatible or were not generated in this study”. We have added this information to the Methods and to the caption of Fig. 3, as suggested by the Reviewer further below.

Scenarios not covered are either incompatible or were not generated with the AIM model (Fujimori et al. 2018).

Line 60: Should it be “adaptation” rather than “adaption”?

The reviewer is correct, and we have corrected this typo.

Lines 123-125: The maps in panels B and C seems to be described in the wrong order in the figure legend.

We apologise for this mistake, and have corrected the order of the panels in the caption.

Line 132: So the proportion of species losing at least half their range increases from 2016 levels even under the best-case scenario (RCP2.6, SSP1)? If so, it would be good to highlight this fact and explain why it happens despite the increase in average range area under the best-case scenario.

We found that this result was an artefactual consequence of the fact that the historical HYDE land use data and the future AIM data had not been harmonised. In our revised version, we have corrected this, thus ensuring that the land use data is indeed continuous in time. The proportion of species losing more than half their range now also decreases in the scenario RCP2.6 / SSP1.

To mirror the results for the historical period in Fig. 1B, in our revised manuscript, Fig. 3 now also shows the proportion of species projected to experience losses of >50% of the natural range size.

Lines 133-134: “increasing levels of global warming result in more species experiencing critical range losses”. More compared with what? More as a function of increasing global warming? This is not very

surprising. It would be nice to make some comparison between impacts of land conversion versus impacts of climate-driven biome shifts.

We have rephrased the sentence referred to by the Reviewer as follows:

Isolating the impact of climate change shows that higher levels of global warming increase both the number of species experiencing substantial range contractions and range expansions^{10,13}.

We feel that Fig. 3, and the accompanying text, along with Figs. S1 and S3, provide sufficient discussion of the impacts of land use and climate change.

Lines 136-139: It is interesting that there was often more variability among SSP scenarios than among RCP scenarios. However, I can't help wondering whether this would still be the case if direct effects of climate change on species distributions had been captured by the models.

The method used in our analysis does not enable us to formally explore this possibility. However, we would not expect the result referred to by the Reviewer to change under an approach that accounts for the impact of changing climatic conditions that too small to manifest as biome changes: A key pattern shown in Fig. S1 is that climate change benefits some species whilst harming others, and that a consequence of these two effects is a relatively small variation in the median range change. Whilst a fully climate-based (as opposed to biome-based) approach may be able to account for range changes as the result of more subtle climatic variations, we do not feel that there is reason to expect that such range changes would be systematic range losses (or gains) in a way that is fundamentally different from the biome-based dynamics and that would have a major impact on the median range change.

In contrast, increased land conversion has systematic negative effects on the vast majority of species, and therefore consistently results in a negative median range change. Given the substantial range of projected land use scenarios (cf. Fig. S4), we consider the increased variability among SSP scenarios compared to RCP scenarios very plausible.

Line 140: It would be helpful to remind readers here what the different SSP scenario trajectories are.

We have added brief summaries of each SSP.

Lines 146-149: It would be helpful in the figure legend to remind readers why some SSP-RCP combinations were not considered. Also, please explain in the legend why uncertainty was not considered in the future projections (assuming there is no way to do this).

We have added the following sentence to the figure caption:

Scenarios not covered are either incompatible or were not generated with the AIM model (Fujimori et al. 2018).

Including additional climate models in our analysis has allowed us to assess the uncertainty of our future projections with respect to the climate data. We have added the following sentence to the caption:

Coloured shades represent standard deviations, and indicate the uncertainty of the projections with respect to the climate data (Methods). Uncertainties of the AIM land use projections of specific SSP-RCP scenarios are not available.

Lines 155-156: Closing yield gaps may cause an increase in biodiversity. However, the land-use intensification that comes with closing yield gaps is typically associated with a reduction of biodiversity. Intensification effects on species were not included in the models here, so it is difficult to say with any confidence whether such a strategy would indeed be beneficial.

We have replaced “closing yield gaps” with “sustainable intensification of production” to emphasise the need for ecologically beneficial approaches. We have added additional references that present evidence of the biodiversity benefits achievable from minimising the need for additional expansion into natural habitat by means of higher-intensity farming.

Lines 203-204: Give some more detail about the spatial upscaling methods used. E.g., what function was used to average values when upscaling?

We have changed the sentence to:

These data were upscaled from their original spatial resolution of 0.083° to a 0.5° grid by summing up the cropland, pasture or urban area of all 0.083° grid cells contained in a given 0.5° cell.

Lines 229-230: More detail is needed describing the bias-correction method for the climate projections.

We have added the following paragraph to summarise the Delta Method:

The delta method is based on applying, in each grid cell, the difference between simulated and observed climate at a point in time at which both are available to the simulated climate at points in time at which only simulated data exist, in order to correct systematic biases in the climate model (Maraun and Widmann, 2017). When the observed data is provided at a higher spatial resolution, the delta method additionally serves to downscale the simulated climate to the resolution of the observations.

Lines 255-256: Give more detail about the species-specific biome types. Which data from Birdlife and IUCN were these exactly.

We have added the following clarification:

Biome requirements include one or more habitat categories (cf. Table S1) in which the species is known to occur. A species was estimated to be present in a grid cell within its extent of occurrence under potential natural vegetation at a given point in time if the species’ list of habitat categories contained the local (i.e. grid cell-specific) potential natural biome at the relevant time. This required matching IUCN habitat categories (<https://www.iucnredlist.org/resources/habitat-classification-scheme>) with the biome categories of the Biome4 vegetation model, which was done as shown in Table S1.

The habitat requirements are available from the websites specified in the text.

Lines 264-265: Give some more detail about the procedures used to convert spatial polygons to grid format.

We have added the following clause:

... using the 'rasterize' function of the 'raster' package in R, which maps spatial polygons to those raster grid cells whose centres are contained within the polygons.

Lines 279-280: The treatment of species land-cover preferences could be clearer here.

We clarified the paragraph as follows:

Each species' actual range at a given time was derived by removing any unsuitable anthropogenic land from the potential natural range. Historical and projected land use maps (data IV; see section above) provide the fraction of each grid cell that is occupied by cropland, pasture or built-up land. These data were combined with information on which of these three artificial land cover types, if any, species can occur in, which is included in the list of species' biome requirements (data II). This allowed us, for each grid cell contained in a species' potential natural range at a given time, to estimate the fraction of the grid cell that contained suitable habitat at that time. A species' actual range size was then obtained as the sum of the areas of suitable habitat from all grid cells.

Lines 292-298: I think the treatment of baselines could be explained better. I think the gist of it is that the baseline 'potential' range is in the absence of human land use and for the distribution of biomes as in 1850. Is that correct?

Yes, the reviewer is correct. We have added the following sentence to the paragraph:

Thus, $A_i^{\text{potential}}(t_0)$ represents the potential natural range size of a species in the reference year 1850, i.e. the range size estimated in the absence of anthropogenic land use and based on the global distribution of biomes in 1850.

Lines 362-363: The date of classification of threat status used in this figure needs to be given, as threat status changes over time.

We have added the date of classification to the figure caption.

Figure S3: It is really difficult to see the changes shown in this figure. I think it would be better to split this figure over multiple pages, so that the individual panels can be larger.

We have split the figure over two pages and increased the size of the panels, as suggested.

Lines 383-386: In the figure legend, explain why the estimates of total agricultural land for SSP5 for the different RCPs start at different levels in 2016.

X-axes of the projected total agricultural land start in 2020. We have clarified this in the caption. Having harmonised the AIM land use projections with the HYDE dataset in our revised version, the estimates of total agricultural land coincide at present-day.

Reviewer 4

This is a fully-revised version of the paper I reviewed for Nature. I applaud the authors for the quality of the revision. I believe it was worth to go for Nat Comm that allows for longer and more detailed papers.

I only have few more comments that should not be too difficult to address:

- I think the authors must discuss that the habitat preferences they used are based on known current requirement/use, that already incorporate human influence on species ranges. Let's take an example with bears or wolves in Europe. If I was an expert, I would believe those species only prefer mountain habitats, sparsely-vegetated habitats etc... this is obviously not true. There are where the humans let them survive. This is likely to be true for hips of species, and I guess this is also very context dependent.

We have added the following paragraph to the text:

Although extents of occurrence are based not only on known, but also inferred and projected occurrences, the data remain very likely biased as the result of range contractions that have occurred before the beginning of the systematic collection and mapping of species' distributions, and that cannot be fully reconstructed. Whilst this may lead us to underestimate the absolute range sizes of species, it does not necessarily imply that we either systematically underestimate or overestimate the percentage change of species' ranges through time.

- While I appreciate the efforts of the authors to defend their modeling strategy, it has to remain fair and not misleading. The whole paragraph around climate niche model (L331-L345) it at best misleading. What is the point of using those papers very specific on few aspects and turn them as strong critics towards species distribution models? There are far more numerous papers demonstrating the usefulness of these models, comparison to independent data, simulated data etc... that show they do work remarkably well... It has been shown that ensemble modeling could address most of the pitfalls highlighted here, and last but not least, most of the caveats listed in this section also applied to the approach used by the authors (extrapolation (seem my previous comment), adequate data for calibration (this applies too here!)).

Following the Reviewer's comment, we have rewritten the paragraph as follows:

Climate niche models estimate statistical relationships between climatic conditions and species' spatial distributions, and apply these to climate projections to estimate future distribution patterns⁶²; as such, they are not subject to the above limitations. By design, they have great potential for mapping species distributions under a high degree of complexity in terms of possible abiotic and biotic predictor variables and their interactions, which has made the approach very useful in scenarios where the number of species, the geographic region and/or the temporal scale considered is relatively small so that statistical challenges are well-manageable⁶³⁻⁶⁵. In an analysis involving the large number of species, points in time, and different scenarios considered here, the challenges commonly faced by climate nice models, specifically in terms of ensuring robustness of the underlying statistical model and the estimated parameters, and avoiding unwanted artefacts in the extrapolation behaviour⁶⁶⁻⁷¹, would be very difficult to manage. By operating directly and transparently on the empirical data of species' extents of occurrence and biome requirements, and not being reliant on any particular statistical model or parameterisation, the approach used here provides the robustness needed at this scale of data^{22,72}.

Reviewers' Comments:

Reviewer #1:

Remarks to the Author:

The authors have again made significant changes to the manuscript that have now made it acceptable for publication in my opinion. Thanks to the authors for their hard work in accommodating my suggestions and those of the other reviewers.

Reviewer #2:

Remarks to the Author:

The authors have generally done a good job in responding to the reviewer comments and the manuscript is substantially improved, particularly with the addition of a consideration of a wide range of future climate and land use scenarios.

I am still concerned, however, about the reliance of some of the main results on the spatial pattern of future biome distribution. In this I see two remaining issues in the methodology:

1. If the simulated biome area stays the same in a future scenario, but the location of the biome moves in space, does the animal habitat area stay the same? This would imply that an animal species is freely capable of migration from its current range to track the spatial shift in biome. How realistic is this assumption? Would it be possible to do a sensitivity test where the geographic ranges of all species are fixed in the period for which they were calibrated (presumably late 20th-early 21st century)? Then, if the biome shifts "out from underneath" the species range, the species is effectively stranded in unsuitable habitat even if suitable habitat might exist elsewhere. I realize that some species (e.g., birds) might find it easy to handle geographic shifts in their range, while others (e.g., large mammals, amphibians) might have a much harder time physically moving to new locations with suitable habitat).

2. How realistic are the rates of change of biomes themselves, particularly in the future? A possible sensitivity test on this would be to perform additional experiments considering only changes in future land use while keeping all biome distributions at their early 21st century spatial pattern.

In a previous review, one reviewer asked about temporal lag in biome distribution change. This was not addressed in the rebuttal document. BIOME4 is an equilibrium vegetation model that simulates the vegetation that would develop in response to the long-term mean climate state on a roughly centennial time scale (it could be longer in colder biomes). Some of the biome shifts presented in Movie S1 are very large and simulated to occur over a 10-year period, for example the very rapid northward expansion of boreal and temperate forests (seen in Movie S1). These rapid spatial shifts are neither supported by paleoecological evidence nor modeling that explicitly accounts for the migrational lag of plants (e.g., Epstein et al., 2007; Nabel et al., 2013). There is little indication that such rapid spatial changes in biome distribution are realistic and it has been noted that most vegetation models have difficulty handling the migrational lag problem, even in the tropics (e.g., Corlett & Westcott, 2013; Scheiter et al., 2020). Although BIOME4 is computationally very efficient, it is incapable of simulating the type of novel plant communities and even biomes that could emerge in the future because of migrational lag and the formation of transient no-analog climates. Presumably these would have a strong effect on realized species ranges over the coming decades (Williams et al., 2007; Stralberg et al., 2009).

The authors therefore need to do something to address the temporal lag problem in their experimental design. One possibility might be to perform additional sensitivity tests that only consider future anthropogenic land use change and consider all biomes fixed in their present-day distribution. It wouldn't be ideal because presumably some biome distributions will change, probably in favor of novel or "no-analog" biomes that represent a transition phase between

present and future climate, and that BIOME4 is incapable of simulating, e.g., species poor grasslands and shrublands occupied by ruderal or even exotic invasive species. Also direct climatic changes (e.g., extremes) may affect animal ranges before the actual biome changes.

Furthermore, it is very difficult to appreciate the changes in the modeled biomes from the way they are presented in the movie. As the changes in biome distribution over time are a critical part of the study, I would prefer to see these results presented in a one or more pages of multi-panel plots showing the range of past and future global biome maps. At the very least, the movie would need to be split into several movies going across the historical and range of future scenarios.

Finally, what was so difficult about using the standard BIOME4 color scheme? The RGB colors could be easily captured from the image on the BIOME4 web page (<https://pmip2.lsce.ipsl.fr/synth/biome4.shtml>) or retrieved through inquiry with the model's authors. The authors' color scheme for biomes is painfully garish.

References

- Corlett, R. T., & Westcott, D. A. (2013). Will plant movements keep up with climate change? *Trends Ecol Evol*, 28(8), 482-488. doi:10.1016/j.tree.2013.04.003
- Epstein, H. E., Yu, Q., Kaplan, J. O., & Lischke, H. (2007). Simulating future changes in Arctic and subarctic vegetation. *Comput Sci Eng*, 9(4), 12-23. doi:10.1109/Mcse.2007.84
- Nabel, J. E. M. S., Zurbriggen, N., & Lischke, H. (2013). Interannual climate variability and population density thresholds can have a substantial impact on simulated tree species' migration. *Ecol Model*, 257, 88-100. doi:10.1016/j.ecolmodel.2013.02.015
- Scheiter, S., Kumar, D., Corlett, R. T., Gaillard, C., Langan, L., Lapuz, R. S., Martens, C., Pfeiffer, M., & Kyle, T. W. (2020). Climate change promotes transitions to tall evergreen vegetation in tropical Asia. *Global Change Biology*, n/a(n/a). doi:10.1111/gcb.15217
- Stralberg, D., Jongsomjit, D., Howell, C. A., Snyder, M. A., Alexander, J. D., Wiens, J. A., & Root, T. L. (2009). Re-shuffling of species with climate disruption: a no-analog future for California birds? *Plos One*, 4(9), e6825. doi:10.1371/journal.pone.0006825
- Williams, J. W., Jackson, S. T., & Kutzbach, J. E. (2007). Projected distributions of novel and disappearing climates by 2100 AD. *Proc Natl Acad Sci U S A*, 104(14), 5738-5742. doi:10.1073/pnas.0606292104

Reviewer 1

The authors have again made significant changes to the manuscript that have now made it acceptable for publication in my opinion. Thanks to the authors for their hard work in accommodating my suggestions and those of the other reviewers.

We are grateful to the Reviewer for their approval of our manuscript, and would like to thank them again for their valuable input.

Reviewer 2

The authors have generally done a good job in responding to the reviewer comments and the manuscript is substantially improved, particularly with the addition of a consideration of a wide range of future climate and land use scenarios.

I am still concerned, however, about the reliance of some of the main results on the spatial pattern of future biome distribution. In this I see two remaining issues in the methodology:

1. If the simulated biome area stays the same in a future scenario, but the location of the biome moves in space, does the animal habitat area stay the same? This would imply that an animal species is freely capable of migration from its current range to track the spatial shift in biome. How realistic is this assumption? Would it be possible to do a sensitivity test where the geographic ranges of all species are fixed in the period for which they were calibrated (presumably late 20th-early 21st century)? Then, if the biome shifts “out from underneath” the species range, the species is effectively stranded in unsuitable habitat even if suitable habitat might exist elsewhere. I realize that some species (e.g., birds) might find it easy to handle geographic shifts in their range, while others (e.g., large mammals, amphibians) might have a much harder time physically moving to new locations with suitable habitat).

We thank the reviewer for providing clear guidance on what they would find a convincing test. We have conducted the sensitivity test suggested by the Reviewer, i.e. we included grid cells in a species’ range only if the species’ habitat requirements included both the vegetation at the relevant time *and* the current vegetation. Results are shown in the newly added Fig. S8, and discussed in the following new paragraph:

As noted in the main text, our estimates of future habitat ranges represent upper estimates of species’ actual geographic distributions. In particular, our main analysis does not account for species’ ability to migrate to areas that will become suitable habitat at a future point in time, but are not at present. However, our framework allows us to examine the effect of excluding such areas from the estimated range. We repeated our analysis of future changes in range sizes, but considered a grid cell as part of a species’ habitat range only if the local biome estimated for both the relevant point in the future and for the present (year 2016) were included in the species’ list of biome requirements. In other words, grid cells outside of species’ current potential natural ranges were not counted towards their future range sizes, assuming that species are not able to migrate at all. This represents an extreme scenario that will underestimate most species’ mobility (e.g. over half of all species considered here can fly) and their ability to track biome shifts. Since the range derived for a species is a subset of the one estimated in our main analysis, projected range losses based on this approach are, by design, higher (Fig. S8). Qualitatively, results are equivalent to those in Fig. 3 in terms of how different RCP/SSP scenarios rank relative to each other.

We have also added a clause to the main text clarifying that our main estimates of habitat ranges do not account for species' dispersal ability.

2. How realistic are the rates of change of biomes themselves, particularly in the future? A possible sensitivity test on this would be to perform additional experiments considering only changes in future land use while keeping all biome distributions at their early 21st century spatial pattern.

In a previous review, one reviewer asked about temporal lag in biome distribution change. This was not addressed in the rebuttal document. BIOME4 is an equilibrium vegetation model that simulates the vegetation that would develop in response to the long-term mean climate state on a roughly centennial time scale (it could be longer in colder biomes). Some of the biome shifts presented in Movie S1 are very large and simulated to occur over a 10-year period, for example the very rapid northward expansion of boreal and temperate forests (seen in Movie S1). These rapid spatial shifts are neither supported by paleoecological evidence nor modeling that explicitly accounts for the migrational lag of plants (e.g., Epstein et al., 2007; Nabel et al., 2013). There is little indication that such rapid spatial changes in biome distribution are realistic and it has been noted that most vegetation models have difficulty handling the migrational lag problem, even in the tropics (e.g., Corlett & Westcott, 2013; Scheiter et al., 2020). Although BIOME4 is computationally very efficient, it is incapable of simulating the type of novel plant communities and even biomes that could emerge in the future because of migrational lag and the formation of transient no-analog climates. Presumably these would have a strong effect on realized species ranges over the coming decades (Williams et al., 2007; Stralberg et al., 2009).

The authors therefore need to do something to address the temporal lag problem in their experimental design. One possibility might be to perform additional sensitivity tests that only consider future anthropogenic land use change and consider all biomes fixed in their present-day distribution. It wouldn't be ideal because presumably some biome distributions will change, probably in favor of novel or "no-analog" biomes that represent a transition phase between present and future climate, and that BIOME4 is incapable of simulating, e.g., species poor grasslands and shrublands occupied by ruderal or even exotic invasive species. Also direct climatic changes (e.g., extremes) may affect animal ranges before the actual biome changes.

Once again, we would like to thank the reviewer for engaging with our manuscript in such a constructive manner and providing a clear test that would allay their concerns. We have conducted the sensitivity test suggested by the Reviewer, i.e. we used the global biome distribution estimated for the year 2016 in our estimation of future range changes, so that range change are computed only in response to projected land use changes. Results are shown in the newly added Fig. S7, and discussed in the following new paragraph:

Estimates of temporal delays in biome shifts in response to climatic changes (Svenning and Sandel, 2013) are currently not available with the global coverage that would allow us to further refine our approach of assuming that biomes at a given point in time are determined by the climatic conditions in the preceding 30 years.

This also applies to data on the dispersal speeds of plant functional types, and their effect on potential delays in colonisations of previously climatically unsuitable areas (Corlett and Westcott, 2013); current studies on this topic are too spatially scarce to inform our approach. In our main analysis, we therefore follow the assumption commonly made in global vegetation models of no seed dispersal limitations (Snell et al., 2014). However, to explore the impact of this assumption, we also repeated our analysis based on the extreme scenario of biomes not shifting at all between the present (year 2016) and 2100. The estimated species' range size changes (Fig. S7) are quantitatively similar to the results of our main analysis (Fig. 3), consistent with our assessment of the overall stronger impact of land use compared to climate-driven biome changes. Qualitatively, i.e. in the terms of how different RCP/SSP scenarios rank relative to each other, results are equivalent to those of our main analysis.

Furthermore, it is very difficult to appreciate the changes in the modeled biomes from the way they are presented in the movie. As the changes in biome distribution over time are a critical part of the study, I would prefer to see these results presented in a one or more pages of multi-panel plots showing the range of past and future global biome maps. At the very least, the movie would need to be split into several movies going across the historical and range of future scenarios.

We appreciate the Reviewers advice on the visualisation of the biome maps. We would point out that for each future point in time, we use a total of 12 climate realisations (4 RCPs \times 3 climate models). Displaying even only a few selected times at a reasonable resolution would therefore likely require 10+ pages. We are sceptical as to whether this format would make it easier for readers to compare global biome maps at different times and for different RCPs and climate models.

Following the Reviewer's suggestion, we have split the supplementary movie into 13 separate movies, displaying the historical period and the 12 projections of the future period.

Finally, what was so difficult about using the standard BIOME4 color scheme? The RGB colors could be easily captured from the image on the BIOME4 web page (<https://pmip2.lscce.ipsl.fr/synth/biome4.shtml>) or retrieved through inquiry with the model's authors. The authors' color scheme for biomes is painfully garish.

We are grateful to the Reviewer for their advice on improving the appearance of the biome maps, and we have changed the colour palette to the BIOME4 scheme, as suggested.